# Pouchitis Is Associated with Paneth Cell Dysfunction and Ameliorated by Exogenous Lysosome in a Rat Model Undergoing Ileal Pouch Anal Anastomosis

**DOI:** 10.3390/microorganisms11122832

**Published:** 2023-11-22

**Authors:** Yi Xu, Zeqian Yu, Song Li, Tenghui Zhang, Feng Zhu, Jianfeng Gong

**Affiliations:** Department of General Surgery, Jinling Hospital, Affiliated Hospital of Medical School, Nanjing University, Nanjing 210093, China; 18260026110@163.com (Y.X.); yuzeqiannju@163.com (Z.Y.); y_lisong@163.com (S.L.); tenghuiff@163.com (T.Z.); zhuf2020@163.com (F.Z.)

**Keywords:** Paneth cell, pouchitis, lysozyme, bacteria dysbiosis

## Abstract

Background: Pouchitis is a common complication of restorative proctocolectomy and ileal pouch anal anastomosis (IPAA) for ulcerative colitis (UC), significantly affecting the postoperative quality of life. Paneth cells play an important role in the maintenance of gut homeostasis. This study aimed to investigate the role of Paneth cells in the pathogenesis of pouchitis. Method: Endoscopic biopsies from the pouch body and terminal ileum of UC patients undergoing IPAA with or without pouchitis were obtained to analyze Paneth cell function. Acute pouchitis was induced with 5% dextran sulfate sodium (DSS) for seven consecutive days in a rat model of IPAA. The Paneth cell morphology was examined by immunofluorescence and electron microscopy. The effect of exogenous lysozyme supplementation on pouchitis was also investigated. The fecal microbiota profile after DSS and lysozyme treatment was determined by 16s rRNA ITS2 sequence analysis. Result: Abnormal mucosal lysozyme expression was observed in patients with pouchitis. The rat model of pouchitis showed increased pouch inflammation, increased CD3+ and CD45+ T cell infiltration, and decreased tight junction proteins, including ZO-1 and Occludin. There is a significant deficiency of Paneth cell-derived lysozyme granules in the rat model of pouchitis. Supplementation with exogenous lysozyme significantly ameliorated pouchitis, lowering the levels of inflammatory cytokines such as TNF-α and IL-6 in the pouch tissue. 16s rRNA analysis revealed a higher Lachnospiraceae level after lysosome treatment. Conclusions: Paneth cell dysfunction is prominent in patients and rat models of pouchitis and may be one of its causes. The decrease in Lachnospiraceae, a characteristic of dysbiosis in pouchitis, could be reserved by lysosome treatment. Lysozyme supplementation shows promise as a novel treatment strategy for pouchitis.

## 1. Introduction

Pouchitis is the most common postoperative complication of ileal pouch anal anastomosis (IPAA) for ulcerative colitis (UC), interfering with the pouch function and the patient’s quality of life. Idiopathic pouch inflammation is still etiologically undetermined. Recent studies suggest that dysbiosis with abnormal immune responses is a key factor in pouchitis [1,2,3]. However, the extensive use of antibiotics and probiotics cannot completely cure pouch inflammation, and 10–15% can still evolve to recurrent or chronic antibiotic-refractory pouchitis [4,5]. Therefore, new therapeutic strategies for pouchitis are required.

The Paneth cell is a secretory lineage lying in the crypts of the small intestinal epithelium [6]. The main components are α-defensin, lysozyme, and phospholipase A2, which inhibit the excessive expansion of the microbiota and maintain gut homeostasis by releasing them into the intestinal lumen [7]. Lysozyme is a β-1,4-N-acetylmuramoylhydrolase that enzymatically processes the glycan skeleton of bacterial cell walls [8]. The constant secretion of lysozyme contributes to the normal colonization of commensal microbiota and the host defense against enteric pathogens [9,10,11]. Paneth cell dysfunction has been implicated in the impairment of the mucosal barrier and in susceptibility to intestinal disorders such as irritable bowel syndrome (IBS) [11], necrotizing enterocolitis (NEC) [12], and Crohn’s disease (CD) [13], all characterized by microbiota dysbiosis.

However, the role of Paneth cells in pouch physiology and pouchitis has not been adequately investigated. We hypothesized that abnormal Paneth cell function, characterized by abnormal lysozyme secretion, triggered microbiota dysbiosis to aggravate pouchitis. This study aimed to explore the role of Paneth cell dysfunction and the subsequent dysbiosis in the pathogenesis of pouchitis.

## 2. Materials and Methods

### 2.1. Ethical Considerations

This study was approved by the Ethics Committee of Jinling Hospital (2021NZKY-011-01). Written informed consent was obtained from all patients. The experimental animal protocols were approved by the Animal Ethics Committee of Jinling Hospital.

### 2.2. Patient Cohorts

Patients with an ileal pouch were prospectively recruited between 1 September and 31 December 2021 from the Inflammatory Bowel Disease Center of Jinling Hospital. The inclusion criteria were as follows: (1) post-colectomy diagnosis of UC; (2) with pouch endoscopy; and (3) ileostomy closure within 10 months. The exclusion criteria were usage of any antibiotics, antifungal agents, bacterial probiotics, or fungal probiotic therapy within 4 weeks before stool sampling. 

The patients’ clinical information was collected during outpatient visits. Mucosal biopsies of the pouch and the terminal ileum were obtained during the pouch endoscopy. A normal pouch was defined based on clinical, endoscopic, or histological criteria. Individual pouch disease activity was scored with the Pouchitis Disease Activity Index (PDAI), and pouchitis was defined as PDAI ≥ 7 [14].

### 2.3. Animal Models

Male Sprague-Dawley rats weighing 350–380 g were housed in a specific pathogen-free animal facility at room temperature with a 12 h light/dark cycle and provided with standard rat chow diet and water. Animal care and experiments were conducted according to international guidelines on animal research and ethics. The animals were fasted for 24 h before surgery, and IPAA was performed according to established protocols [15]. 

After surgery, the model had a 30-day recovery period. The mice were then randomly divided into three groups: control group (no intervention), DSS group, and DSS + Lyso group (*n* = 5 per group). Rats in the DSS group were administered with 5% DSS (relative molecular mass: 36,000–50,000; MP Pharmaceuticals, Thessaloniki, Greece) for 7 consecutive days. Rats in the DSS + Lyso group were additionally given oral gavage with 100 mg/kg lysozyme (L6876; Sigma-Aldrich, Burlington, MA, USA) dissolved in 5% bicarbonate buffer 7 days before DSS administration and continuously throughout the experiment.

### 2.4. Sample Collections

Weight changes and fecal scores were recorded daily. The fecal score was scored according to the previous study [16]. Briefly, a lack of stool was scored as 1, diarrhea as 2, a blob of stool as 3, textured stool as 4, and normal stool as 5. The fecal samples were collected prior to animal sacrifice. For tissue harvesting, the rats were sacrificed under anesthesia on the 7th day after DSS treatment. The pouch and pre-pouch tissues were harvested and washed with ice-cold PBS for further analysis. Part of the tissues were fixed in 10% neutral formalin for histological assessment, and the rest were immediately snap-frozen in liquid nitrogen and stored at −80 °C for subsequent experiments.

### 2.5. Histological Assessment

The tissues were embedded in paraffin and stained with hematoxylin and eosin (H & E). The pouch specimens were assessed as previously described [17]. Erosion was scored as follows: 0, negative; 1, focal erosion; 2, erosion in several regions; 3, extensive erosion. Ulceration was scored as follows: 0, none; 1, focal ulceration of the mucosa in half of the superficial regions; 2, total mucosal ulceration at multiple foci; and 3, extensive mucosal ulceration extending to the muscularis mucosa or beyond. Intra-epithelial inflammation was evaluated by counting the number of lymphocytes in 100 epithelial cells at the tips of the villi. Villous atrophy was scored as follows: 0, none; 1, mild; 2, moderate; or 3, severe with villous flattening. Edema at the lamina propria was evaluated as: 0, none or 1, positive. 

### 2.6. Immunofluorescence Assay

Immunohistochemistry was used to assess the expression of lysozyme (1:500; PAB193Ra01; Wuhan, China), CD3 (1:150; Ab5690; Abcam, Cambridge, UK), CD45 (1:150; Ab10558; Abcam, Cambridge, UK), ZO-1 (1:100; sc-33725; Santa Cruz, CA, USA), and Occludin (1:100; sc-133256; Santa Cruz, CA, USA). The slides were blocked for 30 min at room temperature (RT) in Tris hydroxymethyl aminomethane (25 mM, pH 7.4) or PBS containing 5% BSA. The sections were incubated overnight at 4 °C with primary antibody diluted in blocking buffer. The primary antibodies were detected with FITC-conjugated anti-rabbit antibody (1:1000, Ab-6721, Abcam) diluted in blocking buffer for 60 min at RT. The tissue sections were stored at 4 °C until imaging. The slides were visualized by confocal laser-scanning microscopy (Zeiss LSM510, Jena, Germany) and analyzed by Zeiss Image Examiner software (Zeiss Zen 2012). The immunohistochemical images were analyzed using Image-ProPlus6.0 software to assess optical density.

### 2.7. Western Blot

Protein was extracted using immunoprecipitation assay lysis buffer supplemented with proteinase and phosphatase inhibitors (Roche). The proteins (10 μg/lane) were separated by 10% SDS-PAGE and transferred to PVF membranes. The membranes were blocked with 5% BSA (Biosharp, Beijing, China) for 1 h at RT and were then incubated with specific antibodies, including anti-lysozyme (1:500, PAB193Ra01, Wuhan, China), anti-ZO-1 (1:1000, sc-33725; Santa Cruz, CA, USA), and anti-Occludin (1:1000, sc-133256; Santa Cruz Biotechnology, CA, USA), separately in a covered container at 4 °C overnight. The membranes were incubated with HRP-peroxidase-conjugated secondary antibody for 1 h, and the proteins were visualized using an electrochemiluminescence substrate (Thermo Scientific, Norristown, PA, USA).

### 2.8. Quantitative Real-Time Reverse-Transcription PCR

Total RNA was extracted from the pouch tissues using an RNA Extraction Kit (Sigma-Aldrich). The RNA was denatured in the presence of an oligo dT primer and then reverse transcribed with a cDNA Reverse Transcription Kit (P111/P112; Vazyme Biotech Co., Ltd., Nanjing, China). Lysozyme, IL-17, IL-6, IL-10, INF-γ, and TNF-α mRNA were detected using the SYBR Green qPCR kit (R223-01; Vazyme Biotech Co., Ltd.). The expression of each mRNA was normalized to GAPDH. The data were calculated using the 2-ΔCT method. The primers are listed in Appendix A.

### 2.9. Electron Microscopy

For electron microscopy (EM), the pouch tissue was immersed in 3% glutaraldehyde fixation solution in 0.1 mol/L KH_2_PO_4_ at pH 7.4. Next, the samples were rinsed in 0.01 M PBS (pH = 7.4), fixed in PBS containing 1% starved acid for 2h, and rinsed in phosphate buffer. Dehydration was performed in a graded series of ethanol, followed by infiltration with an embedding agent at 37 °C overnight and polymerization at 60 °C for 48 h. The Leica Ultra-Thin Slicer (LeicAUC7, Leica Microsystems, Wetzlar, Germany) was used for ultrathin sections (60–80 nm thick), and double-staining with uranium lead was used to locate Paneth cells. The tissue sections were examined using a Hitachi electron microscope (HT7700, Hitachi, Tokyo, Japan).

### 2.10. Microbiota DNA Extraction and High Throughput Sequencing

Technical support was from Shanghai Genesky Biotechnology Company (Shanghai, China). The fecal genomic DNA was extracted from fecal samples using the QIAamp^®^ DNA Stool Mini Kit (Qiagen, Hilden, Germany) according to the manufacturer’s protocol. Bacterial genomic DNA was used as a template to amplify the V3–V4 hypervariable region of the 16S rRNA gene using the forward primer (5-CCTACGGGNGGCWGCAG-3) and reverse primer (5-GACTACHVGGGTATCTAATCC-3). Thirty nanograms of genomic DNA samples and the corresponding fusion primers were loaded to PCR, and the corresponding PCR parameters were set for amplification. The PCR products were purified using Agencourt AMPure XP magnetic beads (Beckman Coulter, Brea, CA, USA), dissolved in Elution Buffer, and labeled to complete the database construction. An Agilent 2100 BioAnalyzer (Agilent Technologies, Santa Clara, CA, USA) was used to detect the range and concentration of the fragments in the library. Qualified libraries were sequenced by selecting an appropriate platform (HiSeq/MiSeq) according to the size of the inserted fragments. The tags were stitched into reads based on the overlapping relationship between the reads. Based on the OTU and the species annotation results, the species complexity of the samples and the differences in species between groups were analyzed. OTU representative sequences were taxonomically classified using the Ribosomal Database Project (RDP) Classifier v.2.2, with a minimum confidence threshold of 0.6, and trained on the Green Genes database v201305 using QIIME v1.8.0.

### 2.11. Statistical Analysis

The continuous variables in accordance with normal distribution were reported as mean ± SD, the nonparametric continuous variables were reported as median (interquartile range, IQR), and the categorical variables were reported as frequency and percentage. The Student’s *t*-test was used for intergroup comparisons of continuous variables conforming to normal distribution; otherwise, the Mann–Whitney test was used. A two-sided *p* value < 0.05 was considered statistically significant for all analyses. SPSS software (SPSS 22.0) was used for all analyses.

## 3. Results

### 3.1. Abnormal Mucosal Lysozyme Expression in Patients with Pouchitis

To explore the role of Paneth cells in pouchitis, twelve patients were enrolled (five with a normal pouch and seven with pouchitis). The patient characteristics are summarized in Table 1. Seven patients (58.33%) were male and three (25.00%) had a 3-stage IPAA. The median disease duration before IPAA was 48 months. The median age of the pouch was 12 months. The mean of PDAI was 2.60 in patients with normal pouches and 11.00 in patients with pouchitis. None of the patients had a family history of IBD or had undergone surgery for colorectal carcinoma.

Mucosal inflammation was revealed by H & E staining in the tissues with pouchitis, as described by Shen et al. [18] (Appendix A). The immunofluorescence of the biopsy tissue showed decreased lysozyme-positive cells in the inflamed pouch tissue, which was consistent with the results of the previous study [19] (Figure 1A). Although the pouch tissue had a higher lysozyme mRNA level than the terminal ileum, pouchitis (PS) tissue has a lower lysozyme level compared to a normal pouch (NP) (1.35 ± 0.29 vs. 1.00 ± 0.00, *p =* 0.03; 0.78 ± 0.45 vs. 1.35 ± 0.29, *p =* 0.04, respectively; Figure 1B).

### 3.2. Paneth Cell Dysfunction in a Rat Model of Pouchitis

To explore whether there is a functional impairment of Paneth cells in experimental pouchitis, we used a rat model of DSS-induced pouchitis. Immunofluorescence staining revealed a lower lysozyme expression in the DSS group than that of the control group (Figure 2A). The integrated optical density of cell fluorescence was calculated to estimate the amount of lysozyme in the pouch tissue, indicating that the Paneth cell-derived lysozyme was significantly reduced in the pouches of the DSS group compared with the control group (74.81 ± 16.06 vs. 147.94 ± 26.67, *p =* 0.015; Figure 2B). And compared with the pre-pouch tissues, the pouch tissues presented a lower level of lysozyme expression in the DSS group (Figure 2A,B). There were no significant differences in the expression of lysozyme between the control and the DSS group in the pre-pouch tissue and also no significant differences between the pre-pouch and pouch tissues in the control group (Figure 2A,B).

Electron microscopy showed malformations of the lysozyme secretory granules and the cavitating granules in the Paneth cells (Figure 2C). The proportion of normal particles in each group of five fields was calculated, indicating a lower rate of normal secretory granules in the DSS group compared with the control group (37.20 ± 7.36 vs. 88.00 ± 5.74%, *p <* 0.001) (Figure 2D). In the pouch of the DSS group, lysozyme had decreased, confirmed through Western blotting and qPCR, as compared to the control group (0.05 ± 0.02 vs. 0.21 ± 0.09, *p =* 0.05; 0.35 ± 0.30 vs. 1.0 ± 0.0, *p =* 0.002, respectively; Figure 2E–G).

### 3.3. Oral Lysozyme Ameliorated Pouch Inflammation in a Rat Model of DSS-Induced Pouchitis

Then, we examined whether exogenous lysozymes could ameliorate pouch inflammation. The experimental diagram is shown in Figure 3A. The representative gross anatomy of the pouch is shown in Figure 3B,C. The dynamic changes in body weight, shown in Figure 3D, indicated a higher body weight after lysozyme treatment compared with the DSS treatment alone group.

Compared with the control group, a higher histological score and lower fecal score were observed after DSS administration, which was improved by the lysozyme supplement under DSS treatment and is shown in Figure 3E,F. In the DSS group, the H & E staining (Figure 3G) showed that the villi of the pouch epithelium were blunt, and the mucosa was irregularly arranged with more inflammatory cell infiltration, suggesting that inflammation was present in the pouch body but not in the pre-pouch ileum. After lysozyme supplement under DSS treatment, the pouch inflammation was significantly reduced compared with the DSS group (Figure 3F,G).

### 3.4. Inflammatory Response and Epithelial Barrier after Lysozyme Treatment

Immunofluorescence showed that more CD3+ and CD45+ T cells infiltrated the intestinal epithelium after DSS treatment than that of the control group, mostly in the villi (Figure 4A–C). The DSS + lysozyme group showed less CD45+ infiltration than the DSS group (Figure 4C), but no significant differences in CD3+ infiltration were observed between the DSS group and the DSS + lysozyme group (Figure 4B). The expression of the tight junction proteins Occludin and ZO-1 was marked by immunofluorescence in each group (Figure 4D). Quantification analysis showed decreased Occludin and ZO-1 expression after DSS treatment as compared to the control group, while their expression was increased after lysozyme treatment (Figure 4E–G). 

The levels of inflammatory cytokines were also examined. The expressions of tumor necrosis factor (TNF)-α, interleukin (IL)-6, interferon (IFN)-γ, and interleukin (IL)-17 mRNA in the pouch were all significantly elevated in the DSS group compared with the controls (Figure 4H–K). After lysozyme treatment, the expressions of TNF-α and IL-6 were significantly reduced. IL-10 expression was significantly different between the DSS and the control group, as well as between the DSS + lysozyme and the DSS group (Figure 4L). Specific data for between-group comparisons of all experiments are provided in the Appendix A.

### 3.5. Gut Microbiota Profile after DSS and Lysozyme Treatment

Sequencing of the 16S ribosomal RNA gene tags from the fecal samples was performed to examine alterations in the microbiota profile. The 16S rRNA amplicon did not show any changes in α-diversity among groups. Increased β-diversity was observed in the DSS group compared with the control group (Figure 5A,B). In addition, lysozyme supplementation significantly reduced the bacterial diversity (*p =* 0.030).

Bacteroidetes and Firmicutes are dominant in both cohorts, with a minor prevalence of Fusobacteria and Proteobacteria (Figure 5). At the phylum level, DSS treatment significantly increased the abundance of Proteobacteriax (*p =* 0.002) and Elusimicrobia (*p =* 0.010) compared to the control group, both of which decreased after lysozyme supplement in response to the DSS treatment (*p <* 0.05), whereas the abundance of Firmicutes was significantly increased in the DSS + lysozyme group compared with the DSS group (*p =* 0.001). 

Oral lysozyme supplement contributed to a significant increase in Lachnospiraceae (*p =* 0.030), represented by Dorea, Blautia, and Roseburia (*p =* 0.002; *p* = 0.016; and *p* = 0.018, respectively), which have been shown to be beneficial for intestinal homeostasis. More details of the microbial changes in each group are shown in the Appendix A. 

## 4. Discussion

Although the microbiota profile after IPAA has been extensively studied [20,21,22], the pathogenesis of pouchitis is not completely understood. Current observations suggest that the multi-aspect interplay between dysbiosis and abnormal mucosal immune activation is crucial to pouchitis [23,24]. Whether the alterations in microorganisms in the pouch canal of unusual constructions are the cause or consequence of pouchitis requires further clarification. Paneth cell abnormalities are characteristic [25] of intestinal disorders and are used as predictors of the early recurrence of CD [26]. Decreased antimicrobial peptide secretion by Paneth cells can weaken the mucus barrier, subsequently affecting the intestinal microecology [9,27]. 

Although Paneth cell dysfunction has been observed in the pathology of ileitis, aberrant lysozyme production in pouchitis has not been elucidated. Previous studies have suggested that the ileal pouch has different expressions of antimicrobial peptides [19,28]. This study provides evidence of Paneth cell dysfunction and the downregulation of lysozyme in patients and a relevant animal model of pouchitis for the first time. Due to their active secretory function, Paneth cells are susceptible to endoplasmic reticulum stress (ERS), which activates the unfolded protein response (UPR) [29]. However, if there is extensive ERS in Paneth cells, the signaling typically shifts to a pro-apoptotic state. Notably, Shen et al. reported that increased crypt apoptosis is a feature of autoimmune-associated pouchitis [18]. Our study suggested that the aggravation of pouch inflammation was correlated with lower lysozyme levels in the Paneth cells. Exogenous lysozyme treatment decreased proinflammatory cytokine levels and reversed mucosal barrier impairment. Thus, Paneth cell dysfunction might be a new mechanism of pouchitis, in line with other studies on lysozyme supplementation for treating intestinal inflammation [8,30].

Dysbiosis has also been observed in DSS-induced pouchitis. Studies have consistently suggested a reduction in Lachnospiraceae in pouchitis, which is consistent with our findings [22,31]. This also indicates that enteric lysozyme can reduce the number of Lachnospiraceae, representative producers of the short-chain fatty acids (SCFAs) acetate and butyrate, which can facilitate the development of probiotic bacterial consortia that restore in vivo microbiome functions [32,33]. Further studies focusing on the role of Lachnospiraceae in pouchitis pathogenesis are warranted. Moreover, several studies have shown the effectiveness of fecal microbiota transplantation (FMT) and probiotics in the treatment of pouchitis [34,35], which further highlights the role of microbes in the pathogenesis of pouchitis.

This study has several limitations. First, the sample size is small, and additional prospective studies with a more detailed classification are needed to elucidate the role of Paneth cells in pouchitis. Second, the limitations of the rat model prevented us from examining the role of Paneth cells in pouchitis from a genetic perspective. Other genetically modified models may prompt a more detailed investigation of Paneth cell dysfunction rather than simple drug interventions. Third, we did not examine the changes in the microbiome after lysozyme treatment in normal pouches, which may have caused ambiguity in the current study. Therefore, future studies investigating the role of Paneth cells in pouchitis should include multi-dimensional design schemes. 

Overall, an interaction between the host and microbes with an altered inflammatory immune response exists in the pouch environment. The decreased secretion of lysozymes by Paneth cells might be one of the pathological characteristics of pouchitis, which also leads to a lower abundance of Lachnospiraceae that can be restored by oral lysozyme supplementation. Exploration of a new etiological perspective may be beneficial for patients who do not respond well to traditional antibiotic therapies.

## Figures and Tables

**Figure 1 microorganisms-11-02832-f001:**
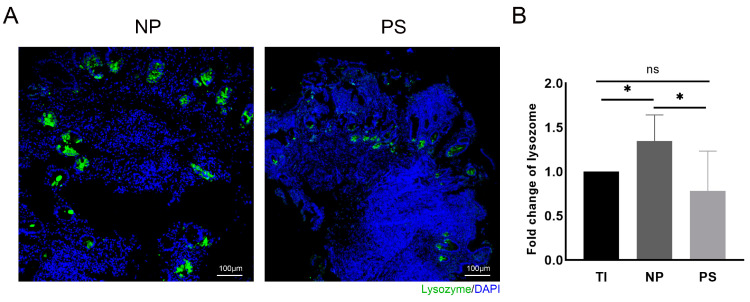
Abnormal lysozyme expression in clinical samples of pouchitis. (**A**) Immunofluorescence assay of lysozyme in pouch tissues (bar = 100 μm); sections stained for indicated markers (Green: lysozyme) and counterstained with DAPI (Blue). (**B**) Relative expression of lysozyme level of biopsy tissues (TI: terminal ileum; NP: normal pouch; PS: pouch with pouchitis). Data are shown as the mean ± SD with 5 samples on TI, 5 samples on NP, and 7 samples on PS. The asterisk indicates a statistically significant difference (* *p* < 0.05, ns: not significant).

**Figure 2 microorganisms-11-02832-f002:**
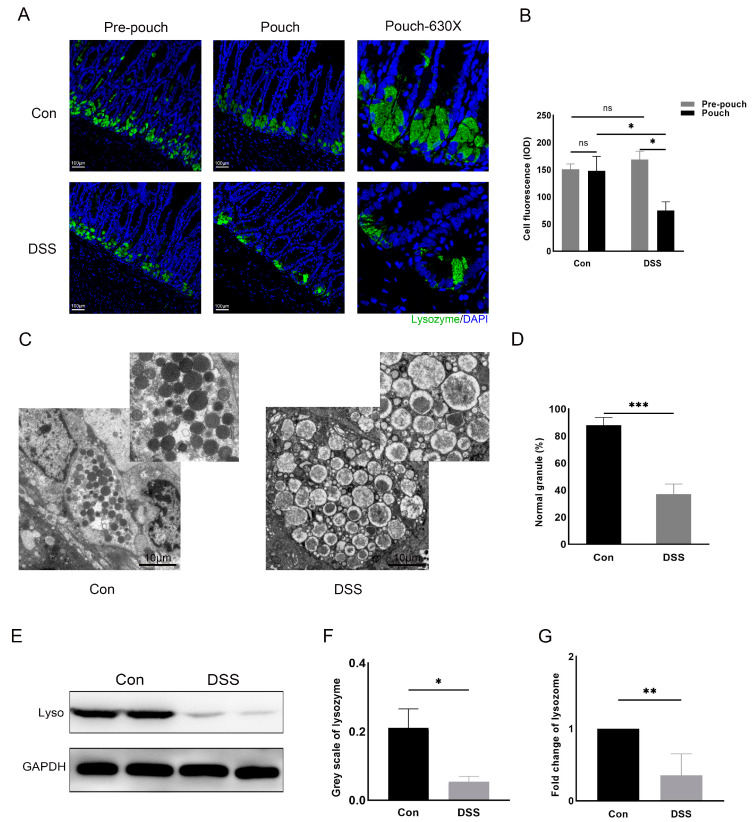
Paneth cell dysfunction in a rat model of pouchitis. (**A**) Immunofluorescence assay of lysozyme in the intestinal tissue of each group (Bar = 100 μm); (**B**) integrated optical density of Paneth cell fluorescence; (**C**) proportion of normal particles in each section; (**D**) electron microscopy revealed normal and contracted appearance of Paneth cells (bar = 10 μm); (**E**) Western blot of the lysozyme protein in different groups; (**F**) greyscale analysis of lysozyme in each group; (**G**) relative expression of lysozyme level in each group. Data shown are representative of 3 independent experiments; mean ± SD derived from 6 rats per group. The marks indicate a statistically significant difference (* *p* < 0.05, ** *p* < 0.01, *** *p* < 0.001, ns: not significant).

**Figure 3 microorganisms-11-02832-f003:**
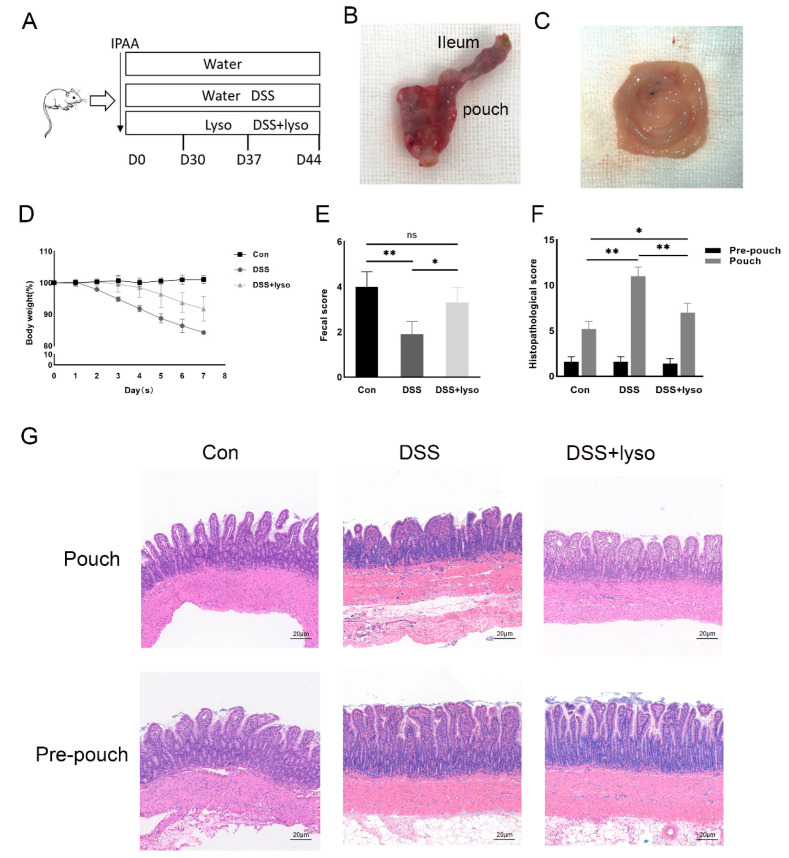
Exogeneous lysozyme ameliorates pouchitis in a rat model of IPAA. (**A**) Schematic diagram of the experiment; (**B**,**C**) pouch configuration and gross pathology; (**D**) change in body weight; (**E**) fecal score; (**F**) histological score; (**G**) H & E staining of pouch tissue in each group (Bar = 20 μm). Data shown are representative of 3 independent experiments; mean ± SD derived from 6 rats per group. The asterisk indicates a statistically significant difference (* *p* < 0.05, ** *p* < 0.01, ns: not significant).

**Figure 4 microorganisms-11-02832-f004:**
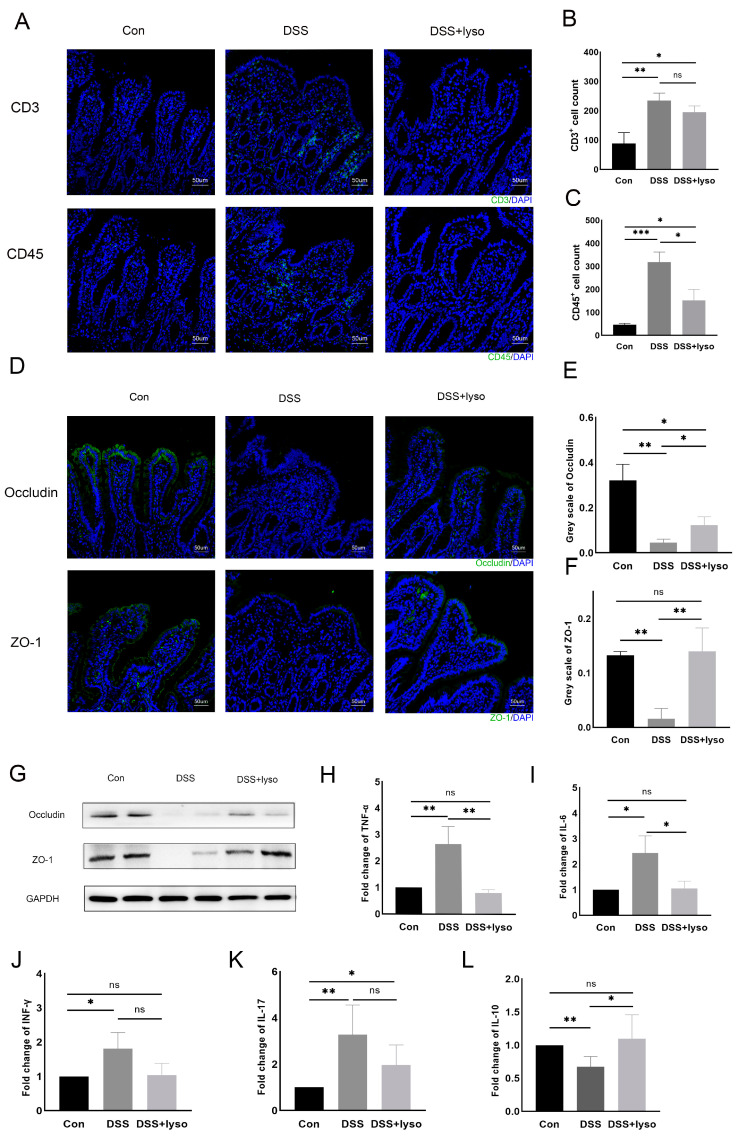
Exogeneous lysosome reduces inflammatory cell infiltration and enhances the mucosal barrier in the rat model of pouchitis. (**A**) Immunofluorescence assay of CD3+ and CD45+ T-cells (Bar = 50 μm). Sections stained for indicated markers (Green: CD3+, CD45+, ZO-1, and Occludin) and counterstained with DAPI (blue); (**B**,**C**) cell count of CD3+ and CD45+ cells per 200× field; (**D**) immunofluorescence assay of Occludin and ZO-1 (Bar = 50 μm); (**E**) greyscale analysis of Occludin; (**F**) greyscale analysis of ZO-1 (Analysis was based on results of 3 independent replicates of the experiment); (**G**) Western blot of Occludin and ZO-1 protein level among groups; (**H**–**L**) relative expression of IL-6, TNF-α, IFN-γ, IL-17, and IL-10 mRNA level among different groups. Data are shown as the mean ± SD derived from 6 rats per group. The asterisk indicates a statistically significant difference (* *p* < 0.05, ** *p* < 0.01, *** *p* < 0.001, ns: not significant).

**Figure 5 microorganisms-11-02832-f005:**
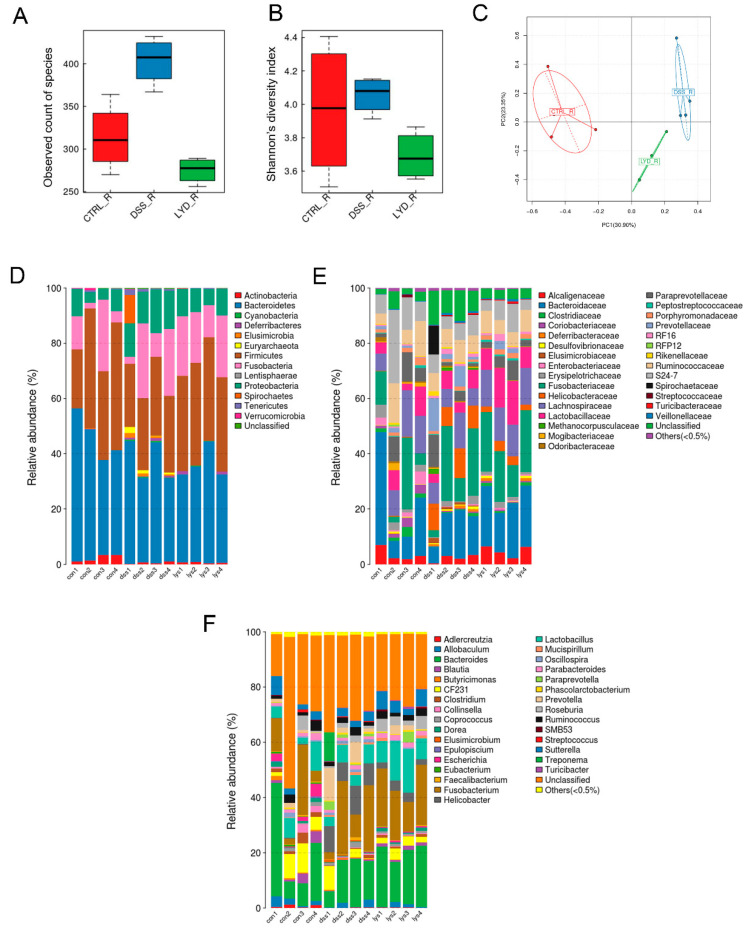
Bacteria biodiversity and composition after DSS and lysozyme treatment in the rat model of pouchitis. (**A**) α-diversity; (**B**) Shannon diversity; (**C**) Simpson diversity; (**D**) relative abundance of bacteria composition at the phylum level; (**E**) relative abundance of bacteria composition at the family level; (**F**) relative abundance of bacteria composition at the genus level (*n* = 4 in each group).

**Table 1 microorganisms-11-02832-t001:** Patient characteristics.

	Normal Pouch (*n* = 5)	Pouchitis (*n* = 7)
Gender (F/M), *n* (%)	2 (40.00)/3 (60.00)	3 (42.86)/4 (57.14)
Age, year, mean ± SD	44.6 ± 9.2	41.6 ± 15.3
Ex-smokers, *n* (%)	1 (20.00)	3 (42.86)
Disease duration, months, median (IQR)	60.00 (24.50–95.25)	36.00 (16.25–57.50)
Pouch age, months, median (IQR)	13 (4.50–17.50)	10.00 (5.00–13.00)
Stage of procedure (II/III), *n* (%)	4 (80.00)/1 (20.00)	5 (71.43)/2 (28.57)
PDAI score, mean ± SD	2.60 ± 0.60	11.00 ± 1.40
Fecal calprotectin, μg/g, mean ± SD	578.9 ± 519.1	935.5 ± 198.5
C-reactive protein, mg/DL, mean ± SD	7.9 ± 6.2	42.1 ± 26.2
Interleukin-6, pg/mL, mean ± SD	7.6 ± 4.3	15.4 ± 8.7
Pre-IPAA treatments		
5-ASA, *n* (%)	5 (100.00)	5(71.43)
Steroids, *n* (%)	4 (80.00)	6 (85.71)
Immunosuppressants, *n* (%)	1 (20.00)	3 (42.86)
Infliximab, *n* (%)	0 (0)	2 (28.57)
Antibiotics, *n* (%)	0 (0)	3 (42.86)
Fecal microbiota transplantation, *n* (%)	0 (0)	2 (28.57)

## Data Availability

Data is contained within the article or Appendix A.

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
