# Peer review of "Pouchitis Is Associated with Paneth Cell Dysfunction and Ameliorated by Exogenous Lysosome in a Rat Model Undergoing Ileal Pouch Anal Anastomosis"

_microorganisms, 2023, doi:10.3390/microorganisms11122832_

Round 1
Reviewer 1 Report
Comments and Suggestions for Authors
Comments to Authors
This study showed that: a) paneth cell dysfunction is prominent in patients and rat models of pouchitis and may be one of its causes; b) the decrease of Lachnospiraceae, a characteristic of dysbiosis in pouchitis, could be reserved by lysosome treatment; c) lysozyme supplementation shows promise as a novel treatment strategy for pouchitis.
Authors are kindly requested to emphasize the current concepts about these issues in the context of recent knowledge and the available literature. This articles should be quoted in the References list.
References
1. Faecal microbiota transplantation (FMT) in the treatment of chronic refractory pouchitis: A systematic review and meta-analysis [published online ahead of print, 2023 Jul 14]. J Crohns Colitis. 2023;jjad120. doi:10.1093/ecco-jcc/jjad120.
2. The efficacy of probiotics on the prevention of pouchitis for patients after ileal pouch-anal anastomosis: A meta-analysis. Technol Health Care. 2023; 31 (2): 401-415. doi:10.3233/THC-220402.
Comments on the Quality of English LanguageMinor editing of English language required
Author Response
Dear Editor,
Thank you very much for taking your time to review our manuscript-Microorganisms-2676882, and providing an opportunity for revising the manuscript. We highly appreciate the reviewers’ and editors’ construction comments and suggestions. As attach, please find the detailed point-to-point response to the reviewers’ comments.
Review #1:
Comment1: Authors are kindly requested to emphasize the current concepts about these issues in the context of recent knowledge and the available literature. This article should be quoted in the References list. References: 1.Faecal microbiota transplantation (FMT) in the treatment of chronic refractory pouchitis: A systematic review and meta-analysis [published online ahead of print, 2023 Jul 14]. J Crohns Colitis. 2023; jjad120. doi:10.1093/ecco-jcc/jjad120; 2. The efficacy of probiotics on the prevention of pouchitis for patients after ileal pouch-anal anastomosis: A meta-analysis. Technol Health Care. 2023; 31 (2):401-415. doi:10.3233/THC-220402.
Response: The above references have been added to the manuscript as follows “Moreover, several studies shown the effectiveness of fecal microbiota transplantation (FMT) and probiotics in the treatment of pouchitis34,35, which further highlights the role of microbes in the pathogenesis of pouchitis.” (Line 324-326)
Comment2: Minor editing of English language required.
Response: We thank the reviewer for this valuable suggestion, and the whole manuscript has been polished accordingly.
Best regards,
Jianfeng Gong
Professor,
Department of General Surgery, Jinling Hospital
Medical School of Nanjing University
Nanjing, PR. China
Reviewer 2 Report
Comments and Suggestions for Authors
Authors used a cross-sectional study of patients and an interventional study of rats to examine an association between pouchitis and paneth cell dysfunction and the effect of exogenous lysosome. However, this article has not fully answered some of the questions due to inadequate statistical analysis and insufficient description.
First, as mentioned by authors, “the sample size is small” (L310), but authors used t-test and ANOVA. These statistical methods are based on assumption of normality, but authors do not justify the assumption. Authors should use appropriate statistical methods or justify what they used for as statistical methods in this manuscript.
Second, authors used “±” and bar as intervals, but authors do not explain what they are (e.g., SD and 95% confidence intervals). Without explanation, it is difficult to judge what authors did in this manuscript. Authors should rewrite the manuscript, carefully.
Third, authors suggest “At the phylum level, DSS treatment significantly increased the abundance of Proteobacteria (p =0.002) and Elusimicrobia (p=0.010), both of which decreased after lysozyme treatment (p<0.05), whereas the abundance of Firmicutes was significantly increased (p=0.001).” (L267), but authors do not show details of these results (e.g., values). Authors showed only p-values, but it is difficult to understand what authors did. Moreover, as mentioned by authors, “we did not examine the changes in the microbiome after lysozyme treatment in normal pouches” (L315), but authors suggest “decreased after lysozyme treatment” and “increased”. Authors should describe manuscript using tables and appropriate words, carefully.
Finally, authors showed a lot of graphs (e.g., Figure 1B and Figure EF), but authors could not explain these results as description in result section. For example, authors showed comparison among TI, NP and PS in Figure 1B, authors do not explain sample of TI (e.g., characteristics and the number of samples). Authors suggest “The integrated optical density of cell fluorescence was calculated to estimate the amount of lysozyme in the pouch tissue, indicating that Paneth cell-derived lysozyme was reduced in the pouches of DSS group compared to the control group (74.81 ± 16.06 vs 147.94 ± 26.67, p=0.015; Figure 2.B)” (L194), but authors do not show any statistical results in Figure 2B. Authors suggest “Decreased lysozyme in the DSS group was also confirmed by Western blotting and qPCR (0.05 ± 0.02 vs 0.21 ± 0.09, p=0.05; 0.35 ± 0.30 vs 1.0 ± 0.0, p=0.002, respectively; Figure 2.E-G).” (L209), but it is difficult to identify which value is for each group. Authors should explain their results and statistical analysis using tables, and explain what authors find in their tables as description in result section, carefully.
Minor comments
Table 1: Authors should ad range of values for variable using median.
Table 1: It is difficult to understand what “smoking history” is (e.g, ever smokers or current smokers).
Author Response
Dear Editor,
Thank you very much for taking your time to review our manuscript-Microorganisms-2676882, and providing an opportunity for revising the manuscript. We highly appreciate the reviewers’ and editors’ construction comments and suggestions. As attach, please find the detailed point-to-point response to the reviewers’ comments.
Review #2: Authors used a cross-sectional study of patients and an interventional study of rats to examine an association between pouchitis and paneth cell dysfunction and the effect of exogenous lysosome. However, this article has not fully answered some of the questions due to inadequate statistical analysis and insufficient description.
Comment1: First, as mentioned by authors, “the sample size is small” (L310), but authors used t-test and ANOVA. These statistical methods are based on assumption of normality, but authors do not justify the assumption. Authors should use appropriate statistical methods or justify what they used for as statistical methods in this manuscript.
Response: Thank you for underling this deficiency. And statistical analysis has been revised as follows: “The continuous variable in accordance with normal distribution were reported as mean±SD, nonparametric continuous variables were reported as median (interquartile range, IQR), and categorical variables were reported as frequency and percentage. Student t test was used for intergroup comparisons of continuous variables conforming to normal distribution, otherwise, Mann-Whitney test was used. A two-sided p value <0.05 was considered statistically significant for all analyses. SPSS software (SPSS 22.0) was used for all analyses.” (Line 163-169)
Comment2: Second, authors used “±” and bar as intervals, but authors do not explain what they are (e.g., SD and 95% confidence intervals). Without explanation, it is difficult to judge what authors did in this manuscript. Authors should rewrite the manuscript, carefully.
Response: According reviewer’s comment, we have thoroughly rewritten this manuscript carefully.
Comment3: Third, authors suggest “At the phylum level, DSS treatment significantly increased the abundance of Proteobacteria (p =0.002) and Elusimicrobia (p=0.010), both of which decreased after lysozyme treatment (p<0.05), whereas the abundance of Firmicutes was significantly increased (p=0.001).” (L267), but authors do not show details of these results (e.g., values). Authors showed only p-values, but it is difficult to understand what authors did. Moreover, as mentioned by authors, “we did not examine the changes in the microbiome after lysozyme treatment in normal pouches” (L315), but authors suggest “decreased after lysozyme treatment” and “increased”. Authors should describe manuscript using tables and appropriate words, carefully.
Response: Thanks for your suggestion. A table showing more details of microbial changes in each group has been added in the manuscript. (Line 290-291, Supplementary Table 3)
Comment4: Finally, authors showed a lot of graphs (e.g., Figure 1B and Figure EF), but authors could not explain these results as description in result section. For example, authors showed comparison among TI, NP and PS in Figure 1B, authors do not explain sample of TI (e.g., characteristics and the number of samples). Authors suggest “The integrated optical density of cell fluorescence was calculated to estimate the amount of lysozyme in the pouch tissue, indicating that Paneth cell-derived lysozyme was reduced in the pouches of DSS group compared to the control group (74.81 ± 16.06 vs 147.94 ± 26.67, p=0.015; Figure 2.B)” (L194), but authors do not show any statistical results in Figure 2B. Authors suggest “Decreased lysozyme in the DSS group was also confirmed by Western blotting and qPCR (0.05 ± 0.02 vs 0.21 ± 0.09, p=0.05; 0.35 ± 0.30 vs 1.0 ± 0.0, p=0.002, respectively; Figure 2.E-G).” (L209), but it is difficult to identify which value is for each group. Authors should explain their results and statistical analysis using tables, and explain what authors find in their tables as description in result section, carefully.
Response: Thanks for your careful checks. (1) The number of samples in each group have been shown in figure legend for all experiments; (2) The new statistical results have been marked in Figure 2.B, and described in detail in the result section (Line 201-205, Figure 2.B); (3) The related description has been thoroughly revised in result section, and a table has been provided to explain our results in this manuscript. (Supplementary Table 2)
Comment5: Minor comments:(1) Table 1: Authors should add range of values for variable using median. (2) Table 1: It is difficult to understand what “smoking history” is (e.g, ever smokers or current smokers).
Response: We have correction according to the reviewer’s comments. (1) Interquartile range (IQR) have added as the range of values for variable using median (Tabel1); (2) The term, “Smoking history”, has been replaced by “History of smoking”.
Best regards,
Jianfeng Gong
Professor,
Department of General Surgery, Jinling Hospital
Medical School of Nanjing University
Nanjing, PR. China
Round 2
Reviewer 2 Report
Comments and Suggestions for Authors
Authors revised the manuscript, but this article has not fully answered some of the questions due to inadequate statistical analysis and insufficient description.
First, authors added a table of figure 3.F in supplementary file, but Figure 3.F has 6 bars. Authors may not include results of pre-pouch in the table. Authors should revise the table of figure 3.F in supplementary file.
Second, authors added supplementary table 3 in supplementary file, but authors do not add the unit of the numbers. Moreover, authors do not add units in Figure 5.A and Figure 5.B. Without explanation of unit, it is difficult for readers to understand what authors did. Authors should add units for the tables and figures.
Third, authors add supplementary figure B, authors did not show the number in T1. Authors should add the number in T1 of figure B.
Finally, authors used the term “History of smoking”, but this term may be inadequate. Authors may suggest that these persons have history of smoking (i.e., ex-smokers), but if so, it may be better to explain it as “ex-smokers”. If they smoke currently, it may be better to explain it as “current smokers”. Authors should use the terms, adequately.
Comments on the Quality of English LanguageAuthors used the term “History of smoking”, but this term may be inadequate. Authors may suggest that these persons have history of smoking (i.e., ex-smokers), but if so, it may be better to explain it as “ex-smokers”. If they smoke currently, it may be better to explain it as “current smokers”. Authors should use the terms, adequately.
Author Response
Review #: Authors revised the manuscript, but this article has not fully answered some of the questions due to inadequate statistical analysis and insufficient description.
Comment1: First, authors added a table of figure 3.F in supplementary file, but Figure 3.F has 6 bars. Authors may not include results of pre-pouch in the table. Authors should revise the table of figure 3.F in supplementary file.
Response: Thank you for underling this deficiency. A revised table of figure 3.F have been provided in supplementary file.
Comment2: Second, authors added supplementary table 3 in supplementary file, but authors do not add the unit of the numbers. Moreover, authors do not add units in Figure 5.A and Figure 5.B. Without explanation of unit, it is difficult for readers to understand what authors did. Authors should add units for the tables and figures.
Response: (1) Supplementary table 3 has been rewritten as Supplementary table 2; (2) The ordinate title of Figure 5.A-B have been rewritten (Figure 5.A: Observed count of species; Figure 5.B: Shannon’s diversity index)
Comment3: Third, authors add supplementary figure B, authors did not show the number in T1. Authors should add the number in T1 of figure B.
Response: The number in T1 of supplementary figure B have been added to the legend of that figure.
Comment4: Finally, authors used the term “History of smoking”, but this term may be inadequate. Authors may suggest that these persons have history of smoking (i.e., ex-smokers), but if so, it may be better to explain it as “ex-smokers”. If they smoke currently, it may be better to explain it as “current smokers”. Authors should use the terms, adequately.
Response: In patient cohort of this study, there were no current smokers. Therefore, the term, “History of smoking” has been replaced by “Ex-smokers” according to the reviewer’s comment.
Best regards,
Jianfeng Gong
Professor,
Department of General Surgery, Jinling Hospital
Medical School of Nanjing University
Nanjing, PR. China